Phylogenomics of darkling beetles (Coleoptera: Tenebrionidae) from the Atacama Desert

Ragionieri Lapo lapo.ragionieri@uni-koeln.de
Zúñiga-Reinoso Álvaro
Bläser Marcel
Predel Reinhard rpredel@uni-koeln.de
University of Cologne, Institute of Zoology , Cologne , Germany
Gillespie Joseph
Electronic publication date: 2023 Feb 23
Publication date: 2023
Volume: 11
Electronic Location ID: e14848
Received 2022 Oct 4; Accepted 2023 Jan 12
Copyright: ©2023 Ragionieri et al.
Copyright year: 2023
Copyright holder: Ragionieri et al.
License: This is an open access article distributed under the terms of the Creative Commons Attribution License, which permits unrestricted use, distribution, reproduction and adaptation in any medium and for any purpose provided that it is properly attributed. For attribution, the original author(s), title, publication source (PeerJ) and either DOI or URL of the article must be cited.
License URL: https://creativecommons.org/licenses/by/4.0/

Keywords: Transcriptome, Neuropeptides, Myosuppressin, Genome, Systematics, Biodiversity, Synapomorphy

Funding: Deutsche Forschungsgemeinschaft 268236062 –SFB 1211 This study was funded by the Deutsche Forschungsgemeinschaft (DFG, German Research Foundation) –Projektnummer 268236062 –SFB 1211. The funders had no role in study design, data collection and analysis, decision to publish, or preparation of the manuscript.

==============================
Background

Tenebrionidae (Insecta: Coleoptera) are a conspicuous component of desert fauna worldwide. In these ecosystems, they are significantly responsible for nutrient cycling and show remarkable morphological and physiological adaptations. Nevertheless, Tenebrionidae colonizing individual deserts have repeatedly emerged from different lineages. The goal of our study was to gain insights into the phylogenetic relationships of the tenebrionid genera from the Atacama Desert and how these taxa are related to the globally distributed Tenebrionidae.

Methods

We used newly generated transcriptome data (47 tribes, 7 of 11 subfamilies) that allowed for a comprehensive phylogenomic analysis of the tenebrionid fauna of this hyperarid desert and fills a gap in our knowledge of the highly diversified Tenebrionidae. We examined two independent data sets known to be suitable for phylogenomic reconstructions. One is based on 35 neuropeptide precursors, the other on 1,742 orthologous genes shared among Coleoptera.

Results

The majority of Atacama genera are placed into three groups, two of which belong to typical South American lineages within the Pimeliinae. While the data support the monophyly of the Physogasterini, Nycteliini and Scotobiini, this does not hold for the Atacama genera of Edrotini, Epitragini, Evaniosomini, Praociini, Stenosini, Thinobatini, and Trilobocarini. A suggested very close relationship of Psammetichus with the Mediterranean Leptoderis also could not be confirmed. We also provide hints regarding the phylogenetic relationships of the Caenocrypticini, which occur both in South America and southern Africa. Apart from the focus on the Tenebrionidae from the Atacama Desert, we found a striking synapomorphy grouping Alleculinae, Blaptinae, Diaperinae, Stenochinae, and several taxa of Tenebrioninae, but not Tenebrio and Tribolium. This character, an insertion in the myosuppressin gene, defines a higher-level monophyletic group within the Tenebrionidae.

Conclusion

Transcriptome data allow a comprehensive phylogenomic analysis of the tenebrionid fauna of the Atacama Desert, which represents one of the seven major endemic tribal areas in the world for Tenebrionidae. Most Atacama genera could be placed in three lineages typical of South America; monophyly is not supported for several tribes based on molecular data, suggesting that a detailed systematic revision of several groups is necessary.

Introduction

Tenebrionidae Latreille, 1802 (Insecta: Coleoptera) have a worldwide distribution and are one of the larger families with more than 30,000 described species (Bouchard et al., 2021). In the majority of species, both larvae and adults are detritivores and often play a significant role in terrestrial food webs (Matthews et al., 2010). Based on their ecological preferences the Tenebrionidae can be broadly divided into two groups: species associated with trees and species with a shift in larval habitat from decaying trees to soil (Matthews et al., 2010). The latter are widely recognized as the insect group best suited for colonizing arid environments and are found worldwide in desert ecosystems. They have developed numerous morphological, physiological and behavioural adaptations to cope with extremely arid conditions and are therefore largely responsible for most of the nutrient cycling in deserts (Cloudsley-Thompson & Chadwick, 1964; Cloudsley-Thompson, 2001; Crawford, 1982; Matthews, 2000; Matthews et al., 2010; Cheli, Bosco & Flores, 2022; Raś, Kamiński & Iwan, 2022). Different from most other insect groups, their biodiversity sometimes increases with aridity (Kergoat et al., 2014a; Koch, 1962; Pfeiffer & Bayannasan, 2012). The genetic basis for these desert adaptations is not yet clear, but it is known that different lineages of the Tenebrionidae have repeatedly migrated into developing deserts in a convergent scenario (Matthews et al., 2010). Currently, 11 subfamilies, 106 tribes and 2,307 genera of Tenebrionidae are recognized (Bouchard et al., 2021), mainly based on the morphological characters (Doyen, 1972; Doyen, 1993; Doyen & Tschinkel, 1982; Kamiński et al., 2020; Matthews et al., 2010; Watt, 1974).

Recent analyses in insect phylogeny resolved the higher-level relationships in many cases using extensive molecular datasets (e.g., Chesters, 2020; Misof et al., 2014; Wipfler et al., 2019). The intra-ordinal relationships in Coleoptera (Bocak et al., 2014; Cai et al., 2022; Gunter et al., 2014; Hunt et al., 2007; McKenna et al., 2019; Zhang et al., 2018) and the intra-familial relationships of the larger beetle families (e.g., Tarasov & Dimitrov, 2016; Nie et al., 2020; Shin et al., 2018; Souza et al., 2020) was also the focus of several such studies. Regarding the Tenebrionidae, unresolved relationships were repeatedly addressed by molecular analyses in recent years, which, among others, consistently confirmed the monophyly of the family (Gunter et al., 2014; Kergoat et al., 2014b; Kamiński et al., 2020). However, these phylogenetic reconstructions are still under discussion because the internal relationships are still not fully solved. In particular, the subfamilies Tenebrioninae Latreille, 1802 and Diaperinae Latreille, 1802 appear to be artificial groups that require thorough revaluation. (e.g., Aalbu et al., 2002; Kergoat et al., 2014b; Kamiński et al., 2020; Johnston et al., 2020). A recent study convincingly suggested the subfamily Blaptinae Leach, 1815 as a monophyletic group based on molecular and morphological analyses (Kamiński et al., 2020); this lineage contains taxa that have traditionally been placed within the presumably polyphyletic subfamily Tenebrioninae. One of the limitations of all of these phylogenetic reconstructions is the lack of comprehensive sampling of lineages from southern Africa and southern South America. Both Southern Africa and South America each have a highly conspicuous tenebrionid fauna including several endemic tribes (e.g., Carrara & Flores , 2015; Koch, 1962; Kuschel, 1969; Matthews et al., 2010; Kamiński et al., 2021) and contain two of the oldest and driest deserts in the world, the Namib and Atacama Deserts (Clarke, 2006; Goudie & Eckardt, 1999) where tenebrionids represent one of the most abundant insect groups.

Aridity in the Atacama Desert can be traced to the Triassic, but the current conditions are closely related to the Andes uplift in the Miocene (Clarke, 2006), because this mountain range acts as an effective rain shadow (Houston & Hartley, 2003). The regions west of the Andes experienced a long-term decrease in precipitation in this context; the corresponding aridification presumably started in the early Miocene in what is now the core area of the Atacama Desert (Dunai, González-López & Juez-Larre, 2005; Ritter et al., 2018) and intensified throughout the Miocene until the present (Jordan et al., 2014; Ritter et al., 2018). Today, the core of the Atacama Desert (Central Depression between 19°S−23°S) is characterized by hyperarid conditions with less than two mm/yr of precipitations (Houston, 2006), making it one of the driest regions on Earth (Clarke, 2006). These climatic conditions are apparently a barrier for the evolution of organisms, and even well-adapted xerophilous insects as darkling beetles avoid the core of the Atacama Desert. Indeed, most tenebrionids prefer peripherally located and slightly wetter habitats in the Coastal and Andean Cordilleras (Fig. 1). However, the long-lasting interactions between tectonic activity and past climate changes in the Atacama Desert created conditions for the diversification of a very peculiar fauna of tenebrionids, some with very ancient relationships (see Endrödy-Younga, 1996; Ferrer, 2015); and under the influence of the fauna of neighboring regions of the Peruvian Desert and the Intermediate Desert of Coquimbo (Peña, 1966a).

Figure 1 Overview of the study area in the Atacama Desert (shaded area).

This region and the adjacent Andean Cordillera are home to about 34 genera of Tenebrionidae, whose phylogenetic relationships are analysed in this study. Also shown are selected representatives of individual genera. Number of Atacama species and total number of species within the genera are noted, respectively. The dotted blue line is the 4,000 m.a.s.l. contour line in the west and the dashed red line is the average annual rainfall isohyet of two mm. The lower panel shows an elevation profile within the study area, exemplified for a cross-section south of Antofagasta (green line) with tenebrionids typical of different elevation levels along this transect. Raster map made with Natural Earth.

The main goals of the current study are obtaining insights (1) into the phylogenetic relationships of the Atacama genera and (2) of the relationships of these taxa to Tenebrionidae from other regions. For this purpose, we collected material for molecular analyses of almost all described tenebrionid genera (30 genera including an undescribed genus of Alleculinae Laporte, 1840) that inhabit the Chilean Atacama Desert including the adjacent Andean Cordillera. Since it is unlikely that analyses of individual genes can resolve all issues concerning the higher phylogeny of the Tenebrionidae, we sequenced transcriptomes of tenebrionid genera from the Chilean Atacama Desert throughout. In addition to the transcriptomes of the Tenebrionidae from the Atacama Desert, the transcriptomes of a larger number of tenebrionid genera from other regions of the world were sequenced to improve taxon sampling for our transcriptome analyses. As a result, our dataset includes seven of the 11 described subfamilies and 47 tribes. We used these data to obtain the deduced amino acid sequences from 35 neuropeptide precursors per species. The suitability of neuropeptide precursor sequences for phylogenetic inferences was previously demonstrated in a proof-of-concept study (Bläser, Misof & Predel, 2020). This approach is relatively fast and simple as it is based on a limited set of easily identifiable and well conserved protein coding genes. In an alternate analysis using the same transcriptome dataset, the rather commonly used approach of compiling a large scale dataset of orthologous genes was performed. Both approaches, the concatenated dataset of neuropeptide precursors and the large scale dataset of orthologous genes were thus used in parallel to evaluate the relationships within the Atacama Tenebrionidae. These analyses resulted in maximum support for most, but not all branches, and enabled a first convincing assessment of the phylogenetic relationships of the Tenebrionidae of the Atacama Desert.

Materials & Methods

Insect collection

Tenebrionid beetles from the Chilean Atacama Desert (30 genera, 14 tribes) were collected by hand between 2017 and 2021 (Table 1; collecting permits CONAF No 005/2017, 105/2020, 016/2021). The collected specimens were either transferred directly into 96% ethanol for DNA and RNA analyses or transported alive for RNA extraction from fresh material; RNA extraction was then carried out in the Cologne laboratory. Furthermore, we collected samples of 51 tenebrionid genera (33 additional tribes) from Central Chile (collecting permits CONAF No 005/2017), Germany, Italy, Spain, Portugal (collecting permit No 757-758/2021/CAPT), Namibia (collecting permit NCRST RPIV01042034) and Peru (collecting permits SERFOR Nr D000019-2022) to improve taxon sampling for phylogenetic analyses. In addition, published peptide precursor sequences of Tribolium castaneum (Herbst, 1797) (Triboliini Gistel, 1848), Zophobas atratus (Fabricius, 1775) (Tenebrionini Latreille, 1802) (Marciniak, Pacholska-Bogalska & Ragionieri, 2022) and Tenebrio molitor Linnaeus, 1758 (Tenebrionini) (Li et al., 2008; Veenstra, 2019; Marciniak, Pacholska-Bogalska & Ragionieri, 2022) were added to our dataset, while peptide precursor sequences of Neomida bicornis (Fabricius, 1777) (Diaperinae: Diaperini Latreille, 1802) were obtained by Blast searches in the NCBI database (https://www.ncbi.nlm.nih.gov/Traces/wgs?val=GDMA01). RNA was additionally extracted from seven taxa of Tenebrionoidea Latreille, 1802 (families Ciidae Leach, 1819, Meloidae Gyllenhaal, 1810, Mycetophagidae Leach, 1815, Pyrochroidae Latreille, 1807, Salpingidae Leach, 1815, Zopheridae Solier, 1834 and one Cleroidea (Melyridae Leach, 1815) (Table 1), which were included in the phylogenetic analyses. Taxonomic determination was carried out by Álvaro Zúñiga-Reinoso and Reinhard Predel.

Table 1 List of Tenebrionidae and outgroup taxa (bold letters) analysed in this study, including statistics of assemblies after filtering.

N50, the largest contigs size at which 50% of bases are contained in contigs of at least this length; BUSCO, Benchmarking Universal Single-Copy Orthologs. TSA, Transcriptome Shotgun Assembly accession number.

Species	Subfamily	Tribe	Country	N50	BUSCOe	TSA	
Achanius piceus	Pimeliinae	Evaniosomini	Chiled	3106	96.2%	GKEL00000000	
Akis trilineata	Pimeliinae	Akidini	Italy	2668	98.0%	GKEQ00000000	
Allecula morio	Alleculinae	Alleculini	Germany	2724	97.5%	GKEV00000000	
Alleculinae gen. n.a	Alleculinae	?	Chiled	1932	96.4%	GKEP00000000	
Alphasida marseuli	Pimeliinae	Asidini	Portugal	1937	97.3%	GKFO00000000	
Alphitobius diaperinus	Tenebrioninae	Alphitobiini	Lab breeding	2028	97.0%	GKFB00000000	
Ammobius rufus	Blaptinae	Opatrini	Portugal	1828	96.8%	GKEO00000000	
Ammophorus cf. peruvianus	Tenebrioninae	Scotobiini	Chiled	2042	96.8%	GKEM00000000	
Antofagapraocis brevipilis	Pimeliinae	Praociini	Chiled	1954	91.8%	GKEN00000000	
Arthroconus sp.	Pimeliinae	Edrotini	Chiled	1489	89.9%	GKHQ00000000	
Aryenis unicolor	Pimeliinae	Evaniosomini	Chiled	1978	92.0%	GKER00000000	
Aspidolobus penai	Pimeliinae	Epitragini	Chile	1903	91.4%	GKET00000000	
Auladera rugicollis	Pimeliinae	Nycteliini	Chile	1813	86.4%	GKFA00000000	
Blaps gibba	Blaptinae	Blaptini	Italy	2401	97.8%	GKHR00000000	
Blapstinus holosericeus	Blaptinae	Opatrini	Chiled	2169	92.1%	GKEU00000000	
Bolitophagus reticulatus	Tenebrioninae	Bolitophagini	Germany	2937	95.3%	GKES00000000	
Caenocrypticoides sp.a	Pimeliinae	Caenocrypticini	Chiled	2373	98.4%	GKFD00000000	
Callyntra unicosta	Pimeliinae	Nycteliini	Chile	1782	86.2%	GKFE00000000	
Cis sp.	Ciidae		Germany	2415	97.9%	GKEY00000000	
Colydium elongatum	Zopheridae		Germany	1798	96.4%	GKHP00000000	
Cordibates chilensis	Pimeliinae	Thinobatini	Chiled	2296	95.5%	GKFF00000000	
Corticeus unicolor	Diaperinae	Hypophlaeini	Germany	2117	96.3%	GKEW00000000	
Cossyphus hoffmannseggi	Lagriinae	Cossyphini	Portugal	1935	97.4%	GKEX00000000	
Crypticus quisquilius	Diaperinae	Crypticini	Germany	2071	96.1%	GKEZ00000000	
Cuphotes mercurius	Stenochiinae	Stenochiini	Chile	1955	90.4%	GKHZ00000000	
Diaperis boleti	Diaperinae	Diaperini	Germany	2181	96.5%	GKFG00000000	
Diastoleus costalenis	Tenebrioninae	Scotobiini	Chiled	2000	95.9%	GKFH00000000	
Dichillus subcostatus	Pimeliinae	Stenosini	Portugal	1778	97.0%	GKFI00000000	
Discopleurus sp.a	Pimeliinae	Stenosini	Chiled	1561	95.6%	GKFJ00000000	
Eledona agricola	Tenebrioninae	Bolitophagini	Germany	2443	97.0%	GKFC00000000	
Entomochilus rugosus	Pimeliinae	Physogasterini	Chiled	2107	83.9%	GKFK00000000	
Eremoecus sp.	Pimeliinae	Trilobocarini	Chiled	2081	95.0%	GKFL00000000	
Erodius goryi obtusus	Pimeliinae	Erodiini	Portugal	2199	96.1%	GKFM00000000	
Eurychora sp.	Pimeliinae	Adelostomini	Namibia	1674	91.0%	GKFN00000000	
Evaniosomus sp.	Pimeliinae	Evaniosomini	Peru	1649	87.8%	GKHH00000000	
Falsopraocis australis	Pimeliinae	Praociini	Chile	2388	93.4%	GKHO00000000	
Geoborus rugipennis	Pimeliinae	Epitragini	Chiled	2522	93.6%	GKHN00000000	
Gonopus sp.	Blaptinae	Platynotini	Namibia	1861	93.8%	GKHM00000000	
Gyrasida camilae	Pimeliinae	Praociini	Chile	1978	96.3%	GKHL00000000	
Gyriosomus curtisi	Pimeliinae	Nycteliini	Chiled	2279	93.2%	GKHJ00000000	
Heliofugus sp.	Stenochiinae	Cnodalonini	Chile	2063	97.6%	GKFX00000000	
Heliotaurus ruficollis	Alleculinae	Cteniopodini	Portugal	1877	85.1%	GKGC00000000	
Hexagonochilus tuberculatus	Pimeliinae	Stenosini	Chile	1988	96.2%	GKFW00000000	
Hylithus cf. tentyroides	Pimeliinae	Edrotini	Chiled	1971	94.3%	GKHK00000000	
Imatismus sp.	Pimeliinae	Tentyriini	Namibia	1305	85.4%	GKGE00000000	
Isomira semiflava	Alleculinae	Gonoderini	Germany	2385	90.0%	GKGA00000000	
Lagria sp.	Lagriinae	Lagriini	South Africa	2023	96.1%	GKGD00000000	
Leptoderis collaris	Pimeliinae	Elenophorini	Spain	2697	95.0%	GKFZ00000000	
Melanimon tibiale	Tenebrioninae	Melanimini	Portugal	2516	97.4%	GKGZ00000000	
Melaphorus elegans	Pimeliinae	Evaniosomini	Chiled	1224	77.3%	GKFP00000000	
Meloe proscarabaeus	Meloidae		Germany	2036	95.4%	GKHC00000000	
Melyris sp.	Melyridae		South Africa	1372	85.9%	GKHD00000000	
Misolampus gibbulus	Stenochiinae	Cnodalonini	Portugal	2211	96.5%	GKHA00000000	
Mycetophagus quadripustulatus	Mycetophagidae		Germany	2623	98.6%	GKHB00000000	
Nalassus laevioctostriatus	Tenebrioninae	Helopini	Germany	2007	96.6%	GKGW00000000	
Neoisocerus ferrugineus	Blaptinae	Dendarini	Portugal	1845	86.7%	GKGY00000000	
Neomida bicornisb	Diaperinae	Diaperini	USA	n/a	n/a	GDMA01.1	
Nestorinus sp.a	Stenochiinae	?	Chile	1924	94.6%	GKGX00000000	
Nyctelia varipes	Pimeliinae	Nycteliini	Chile	1636	88.0%	GKGU00000000	
Nycterinus atacamensis	Tenebrioninae	incertae sedis	Chiled	2120	70.6%	GKGS00000000	
Nyctopetus tenebrioides	Pimeliinae	Epitragini	Chile	2453	96.9%	GKGV00000000	
Omophlus lepturoides	Alleculinae	Omophlini	Germany	2885	96.7%	GKGQ00000000	
Onymacris rugatipennis	Pimeliinae	Adesmiini	Namibia	2079	96.2%	GKHS00000000	
Oochrotus unicolor	Diaperinae	Crypticini	Portugal	1848	96.6%	GKGT00000000	
Opatrum sabulosum	Blaptinae	Opatrini	Germany	1947	96.6%	GKGR00000000	
Pedinus sp.	Blaptinae	Pedinini	Portugal	1986	95.3%	GKHE00000000	
Phaleria gayi	Diaperinae	Phaleriini	Chiled	1840	96.0%	GKGP00000000	
Philorea sp.	Pimeliinae	Physogasterini	Chiled	1969	92.6%	GKGO00000000	
Physogaster sp.a	Pimeliinae	Physogasterini	Chiled	2290	83.5%	GKGM00000000	
Pilobalia sp.a	Pimeliinae	Nycteliini	Chiled	2041	96.3%	GKGK00000000	
Pimelia rugulosa	Pimeliinae	Pimeliini	Italy	1440	90.7%	GKGN00000000	
Platydema violacea	Diaperinae	Diaperini	Germany	2405	97.3%	GKGL00000000	
Praocis sp.	Pimeliinae	Praociini	Chiled	2123	91.5%	GKGH00000000	
Prionychus melanarius	Alleculinae	Alleculini	Germany	2187	96.1%	GKFQ00000000	
Psammetichus pilipes	Pimeliinae	Elenophorini	Chiled	2029	95.5%	GKGI00000000	
Psectrascelis confinis	Pimeliinae	Nycteliini	Chiled	2414	97.6%	GKFR00000000	
Pyrochroa serraticornis	Pyrochroidae		Germany	2805	95.2%	GKFU00000000	
Salax lacordairei	Pimeliinae	Trilobocarini	Chiled	2179	95.1%	GKFV00000000	
Scaurus uncinus	Tenebrioninae	Scaurini	Portugal	2003	95.7%	GKFS00000000	
Scotobius brevipes	Tenebrioninae	Scotobiini	Chiled	2108	87.3%	GKFT00000000	
Sepidium bidentatum	Pimeliinae	Sepidiini	Portugal	1780	96.3%	GKGJ00000000	
Synchita undata	Zopheridae		Germany	2632	96.1%	GKGF00000000	
Tenebrio molitorc	Tenebrioninae	Tenebrionini	Lab breeding	n/a	n/a	GIPG00000000	
Tentyria cf. laevigata	Pimeliinae	Tentyriini	Italy	2153	97.3%	GKGG00000000	
Thinobatis calderana	Pimeliinae	Thinobatini	Chiled	2415	96.5%	GKHI00000000	
Tribolium castaneum	Tenebrioninae	Triboliini	Lab breeding	n/a	n/a	GCA_000002335.3	
Trilobocara ciliata	Pimeliinae	Trilobocarini	Chiled	1818	90.1%	GKHG00000000	
Valdivium sp.a	Lagriinae	Adeliini	Chile	1592	95.8%	GKGB00000000	
Vincenzellus ruficollis	Salpingidae		Germany	3049	98.1%	GKFY00000000	
Zophobas atratusc	Tenebrioninae	Tenebrionini	Lab breeding	n/a	n/a	GIPJ00000000	
Zophosis sp.	Pimeliinae	Zophosini	Namibia	1729	92.7%	GKHF00000000	
Notes.

a undescribed species.

b transcriptome data from McKenna et al. (2019).

c transcriptome data from Marciniak, Pacholska-Bogalska & Ragionieri (2022).

d species from Atacama Desert.

e Insecta database (https://busco-archive.ezlab.org/; Select Eukaryota sets, then Metazoa sets, and then Insecta odb9).

RNA extraction, cDNA library preparation and sequencing

Total RNA was extracted from samples stored in absolute ethanol or from individuals kept alive until tissue dissection. To avoid excessive RNA degradation in specimens stored in ethanol, head and pronotum of the beetles were separated from the rest of the body before transferring them into ethanol. In larger species, the body was additionally opened longitudinally with sterilized scissors. Without any treatment prior to storage in ethanol, the RNA was usually highly degraded, suggesting limited penetration of ethanol across the cuticle. Grinding of whole insects was avoided in order to enable the intestine to be removed later. Insects alive until tissue dissection were kept at 4 °C for 10 min before preparation. In most individuals (both ethanol and fresh material), after removal of the appendages (legs, elytra, antennae), the body was opened dorsally with sterilized scissors, the intestine was removed and the central nervous system (CNS) was carefully dissected. In small species, representing the genera Ammobius Guérin-Méneville, 1844, Achanius Erichson, 1847, Colydium Fabricius, 1792, Cordibates Kulzer, 1956, Corticeus Piller & Mitterpacher, 1783, Dichillus Jacquelin du Val, 1861, Discopleurus Lacordaire, 1859, Eledona Latreille, 1796, Melanimon Steven, 1829, Oochrotus Lucas, 1852, Synchita Hellwig, 1792, and Thinobatis Eschscholtz, 1831, the CNS was not dissected. For all other samples, total RNA was extracted from CNS and remaining tissues separately using one mL of TRIzol (Thermo Fisher Scientific, Darmstadt, Germany) following the manufacturers recommendations. Total RNA from each sample was quantified using Qubit RNA Assay Kit (Thermo Fisher Scientific) and subsequently subjected to quality control and RNA integrity number (RIN) as implemented in the Agilent 2100 Bioanalyzer system (Agilent Technologies, Waldbronn, Germany). Finally, RNA from CNS and remaining tissue from each sample were pooled together in equimolar concentrations for library preparations. This approach improved the detection of peptide precursor sequences, whose genes are mainly expressed in the CNS. Sequencing libraries (double-indexed) were prepared using 1 µg of total RNA with the Illumina® TruSeq® stranded RNA sample preparation kit (Cat.20020594; Illumina, San Diego, CA, U.S.A.). If the total RNA concentration was insufficient for standard library preparation, at least 2 ng of extract was pre-amplified using the Ovation RNA-Seq System V2 (NuGen, San Carlos, CA, USA). The library preparation of pre-amplified samples was performed according to the Nextera XT DNA sample preparation protocol (part no. 15031942 Rev. C). Subsequent sample preparation and sequencing was carried out at the Cologne Center for Genomics on an Illumina HiSeq 4000 and Illumina NovaSeq 6000 systems as described in Ragionieri & Predel (2020) with 75 bp or 100 bp paired end reads.

Transcriptome assembly, evaluation of cross-contaminations and statistics

Raw data (FASTQ files format) were filtered by removing adapter sequences and low quality using Trimmomatic 0.38 (Bolger, Lohse & Usadel, 2014). The resulting filtered RAW reads were submitted to NCBI (BioProject: PRJNA884860 Sequence Read Archives (SRA): SRR22314233, SRR22314232, SRR22314230, SRR22314229, SRR22314228, SRR22314227, SRR22314226, SRR22314224, SRR22314223, SRR22314225, SRR22314212, SRR22314210, SRR22314208, SRR22314209, SRR22314222, SRR22314221, SRR22314220, SRR22314218, SRR22314217, SRR22314216, SRR22314214, SRR22314213, SRR22314207, SRR22314215, SRR22314206, SRR22314211, SRR22314219, SRR22314193, SRR22314192, SRR22314191, SRR22314199, SRR22314205, SRR22314204, SRR22314203, SRR22314202, SRR22314201, SRR22314200, SRR22314198, SRR22314197, SRR22314196, SRR22314194, SRR22314195, SRR22314190, SRR22314188, SRR22314187, SRR22314186, SRR22314184, SRR22314183, SRR22314182, SRR22314181, SRR22314180, SRR22314189, SRR22314178, SRR22314177, SRR22314185, SRR22314179, SRR22314172, SRR22314170, SRR22314168, SRR22314175, SRR22314174, SRR22314173, SRR22314167, SRR22314166, SRR22314176, SRR22314164, SRR22314163, SRR22314162, SRR22314161, SRR22314160, SRR22314169, SRR22314159, SRR22314171, SRR22314165, SRR22314158, SRR22314157, SRR22314156, SRR22314154, SRR22314153, SRR22314152, SRR22314151, SRR22314150, SRR22314148, SRR22314147, SRR22314146, SRR22314145, SRR22314155, SRR22314149, SRR22314144, SRR22314143, SRR22314142, SRR22314231). Filtered reads were de novo assembled using Trinity v2.2.0 (Grabherr et al., 2011; Haas et al., 2013) with the read normalization option. All transcriptome assemblies were checked for potential cross-contaminations due to multiplex sequencing of several libraries using CroCo v1.1 (Simion et al., 2018) which removes potential sources of contamination using both transcriptome assemblies and the corresponding paired raw data (Table S1). This strategy uses sequence similarities and abundances to detect potential cross-contaminations. For closely related species that are analysed together, this can lead to an overestimation of cross-contamination (Simion et al., 2018). CroCo was run with the following settings: fold-threshold 2, minimum-coverage 0.1, overexp FLOAT 300, minimum percent identity between two transcripts to suspect across contamination 98%, minimum length of an alignment between two transcripts to suspect a cross contamination 180. Finally, we checked for and eliminated additional contamination of vector and linker/adapter using UniVec database (http://www.ncbi.nlm.nih.gov/tools/vecscreen/univec/). The transcriptomes assembled in this study lost on average about 2% of their sequence information due to the cross-contamination check. The quality and the completeness according to conserved single-copy ortholog content of transcriptome assemblies were evaluated using the Perl script (TrinityStats.pl) included in Trinity and BUSCO v3 based on an Endopterygota obd9 dataset (Seppey, Manni & Zdobnov, 2019), respectively. The filtered transcriptome assemblies were submitted to NCBI Transcriptome Shotgun Assembly database (Table 1) and used for the large scale data set phylogenetic reconstruction.

Orthology assessment and alignment of neuropeptide precursors

Available amino acid sequences of neuropeptide precursors of Tr. castaneum and Te. molitor (Li et al., 2008; Veenstra, 2019; Marciniak, Pacholska-Bogalska & Ragionieri, 2022) were used as initial queries to search for orthologous sequences in the transcriptome assemblies. The assembled transcripts were analysed with the tblastn algorithms provided by NCBI (https://blast.ncbi.nlm.nih.gov/Blast.cgi) or BioEdit version 7.0.5.3 (Hall, 1999). In case of missing data, precursor sequences of closely related taxa were used as alternative query sequences. Candidate nucleotide precursor gene sequences were translated into amino acid sequences using the ExPASy Translate tool (Artimo et al., 2012; http://web.expasy.org/translate/) with the standard genetic code. Orthologous neuropeptide precursor sequences were aligned using the MAFFT-L-INS-i algorithm (Katoh & Standley, 2013) (dvtditr (amino acid) Version 7.299b alg=A, model=BLOSUM62, 1.53, −0.00, −0.00, noshift, amax =0.0); terminal sequences which were only found in few species were manually trimmed. The results were then manually checked for misaligned sequences using, e.g., N-termini of signal peptides and conserved amino-acid residues (cleavage signals, Cys as target for disulfide bridges) as anchor points. Individual amino acid alignments of each group of orthologous neuropeptide precursors were concatenated with catsequences 1.3 (https://zenodo.org/record/4409153#.YmJYT35Byot). The average evolutionary divergence for each neuropeptide precursor was calculated as in Bläser & Predel (2020). Briefly, overall mean distances (± standard error after 500 bootstrap generations) were computed with MEGA X (Kumar et al., 2018) implementing the Poisson correction model (Zuckerkandl & Pauling, 1965). Amino acid compositions and parsimony informative sites of the combined alignment were calculated using MEGA X.

Compilation of an orthologous gene dataset of Tenebrionidae

A Coleoptera orthologous reference gene set was compiled using OrthoDB v10. This approach provides reliable markers for phylogenomics (Misof et al., 2014; McKenna et al., 2019). Single copy genes shared across species of Coleoptera (Taxonomy ID: 7041) were selected for analysis. Orthograph (Petersen et al., 2017) was used to generate a profile hidden Markov model from the amino acid sequences of transcripts of each reference gene on the filtered transcriptome assemblies. Initially, we obtained 2,689 orthogroups (OGs) shared among Coleoptera, which were subsequently aligned using the MAFFT-L-INS-I algorithm (Katoh & Standley, 2013). Alignment ambiguities or spurious sequences in each OG were identified and removed using trimAL 1.2 (Capella-Gutierrez, Silla-Martinez & Gabaldon, 2009) with residue overlap threshold (-resoverlap 0.75) and sequence overlap threshold (-seqoverlap 90). With that approach, 947 out of 2,689 OGs were removed from the initial data set. Finally, all OGs were concatenated in a single partitioned super-alignment using catsequences.

Genome sequencing, assembly and identification of myosuppressin genes

Whole genome extraction was carried out using thoracic muscles of a single individual of Nycterinus abdominalis Eschscholtz, 1829 collected in Talcahuano, Chile. High molecular weight genomic DNA was purified using MagAttract® HMW DNA Kit (Ref. 67563, QIAGEN GmbH, Hilden, Germany). DNA concentration was determined using Qubit 2.0 Fluorometer (Thermo Fisher Scientific). Fragment size was verified using DNA integrity number as implemented in the Agilent 2100 Bioanalyzer system. Genomic DNA library was prepared using the Illumina TruSeq Nano DNA High Throughput Library Prep Kit (Illumina, Cat. No 20015965) with modifications of the protocol (TruSeq DNA Nano Reference Guide, Document # 1000000040135 v00, October 2017). Only one cycle of polymerase chain reaction (PCR) was conducted to complete adapter structures in order to avoid PCR bias. Library validation and quantification were carried out as implemented in Agilent TapeStation, and subsequently the library was pooled and quantified using the Peqlab KAPA Library Quantification Kit (Roche Sequencing Solutions, Inc., USA; KK4835-07960204001) on an Applied Biosystems 7900HT Sequence Detection System and finally sequenced on an Illumina NovaSeq 6000 sequencer with 150 bp paired end reads. Raw data (FASTQ files format) were filtered by removing adapter sequences and low quality reads using Trimmomatic 0.38 (Bolger, Lohse & Usadel, 2014). Filtered raw data were assembled using the programs SOAPdenovo2 (Luo et al., 2012) using different k-mer values. The myosuppressin precursor was identified as described above (2.4). Genomic nucleotide sequences containing introns were subsequently aligned manually in BioEdit version 7.0.5.3 (Hall, 1999).

Phylogenetic analysis of neuropeptide precursors and a large scale orthologous gene dataset

FASTA files of aligned peptide precursor sequences were converted into PHYLIP and NEXUS formats using AliView 1.18-beta7 (Larsson, 2014). After defining the N-terminus of each neuropeptide precursor as starting partition, best-fit partitioning schemes and substitution models for subsequent phylogenetic analyses were predicted with ModelFinder (Chernomor, von Haeseler & Minh, 2016; Kalyaanamoorthy et al., 2017; Minh et al., 2021) implemented in IQ-TREE release 2.1.4b (Minh et al., 2020). Models and concatenated alignments for all analyses of both data sets are listed in Data S1 and S2. All phylogenetic analyses have been rooted using the Cleroidea Melyris sp. Bayesian inference (BI) analyses were run with MrBayes, with four runs, using eight chains and a sample frequency of 1,000 until convergence was achieved (PSFR value between 1.00–1.02) with a 10,000,000 generations (Ronquist et al., 2012). Maximum likelihood (ML) analyses were carried out using IQ-TREE 2.1.4b. ML analyses of both data sets were ran with the nearest-neighbour interchange search to consider all possible nearest-neighbour interchanges (-allnni) and branch support was evaluated with 1,000 ultra-fast bootstrap (UFBoot) (Hoang et al., 2017) and the Shimodaira–Hasegawa-like approximate likelihood ratio test (SH-aLRT) (Guindon et al., 2010). Trees were visualized using FigTree 1.4.2 (http://tree.bio.ed.ac.uk/) and designed in Inkscape 1.0 (https://inkscape.org/).

Results

About 34 native genera of Tenebrionidae were described from the Chilean Atacama Desert (Ferrú & Elgueta, 2011; Peña, 1966a; Vidal & Guerrero, 2007); the exact number depends on the definition of the boundaries of the Atacama (see Fig. 1). We collected and analysed specimens of 30 genera (Epipedonota Solier, 1836, Conibius LeConte, 1851 and Parepitragus Casey, 1907 missing), including genera that inhabit only peripheral regions such as the high Andes (Antofagapraocis Flores, 2000, Pilobalia Burmeister, 1875 and an undescribed genus of Alleculinae) or the salty beaches and dunes of the Pacific coast (Phaleria Latreille, 1802, Thinobatis). In addition, sequence data of introduced species were obtained either from publicly available databases (Te. molitor, Tr. castaneum) or the beetles were sequenced from breeding strains (Alphitobius diaperinus (Panzer, 1797)). The analysed taxa from the Atacama Desert are currently classified in five subfamilies (Alleculinae, Blaptinae, Diaperinae, Pimeliinae Latreille, 1802, Tenebrioninae) and 17 tribes (Table 1). For an assessment of the phylogenetic position of the Atacama genera, we additionally generated a transcriptome dataset encompassing diverse tenebrionid taxa (altogether seven of the 11 described subfamilies, 47 tribes) from other regions of the world (Table 1), taxa of different families belonging to the superfamily Tenebrionoidea (Ciidae, Meloidae, Mycetophagidae, Pyrochroidae, Salpingidae, Zopheridae), and Melyridae.

(A) Analysis of neuropeptide precursors

The primary matrix comprises 6,457 amino acids from 35 neuropeptide and neuropeptide-like precursors; information on sequence length and sequence coverage is provided in Table S2. The average evolutionary divergence over all sequences of the precursor dataset is 0.25 (± 0.03) and differs considerably between the different precursors (Table S2). The best fitting models according to ModelFinder are listed for each partition in Data S1, which also contains the concatenated alignment.

The phylogenetic tree of the concatenated neuropeptide precursor dataset (Fig. 2) recovered Tenebrionidae as monophyletic. Tenebrionidae are separated into one clade containing all the Pimeliinae analysed and a second clade containing the taxa of Alleculinae, Blaptinae, Diaperinae, Lagriinae Latreille, 1825, Stenochinae Kirby, 1837, and Tenebrioninae. All Atacama genera of Pimeliinae belong to a clade with worldwide distribution which is recovered as sister to Akis Herbst, 1799 (Akidini Billberg, 1820) and Pimelia Fabricius, 1775 (Pimeliini Latreille, 1802) from the Mediterranean region. The Atacama Pimeliinae are separated into a lineage containing South American genera of Elenophorini Solier, 1837, Nycteliini Solier, 1834, Physogasterini (Lacordaire, 1859), Praociini Eschscholtz, 1829, and Stenosini Schaum, 1859 and a second lineage including all remaining Pimeliinae from the Atacama Desert. Both clades also include taxa from other regions. Internal branches of the first clade generally show high support. This clade is first divided into a group containing Stenosini and a group containing the Elenophorini, Nycteliini, Physogasterini, and Praociini from the Atacama Desert. Within the Stenosini with Chilean species of Discopleurus and Hexagonochilus Solier, 1851 nests Mediterranean Leptoderis Billberg, 1820 (Elenophorini) as sister to Mediterranean Dichillus (Stenosini). The only described member of the Elenophorini from the Atacama region, Psammetichus Latreille, 1829, appears on a branch with the southern African Eurychora Thunberg, 1789 (Adelostomini Solier, 1834) and Mediterranean Sepidium Fabricius, 1775 (Sepidiini Eschscholz, 1829). The remaining taxa of this large clade branch into Mediterranean Alphasida Escalera, 1905 (Asidini Fleming, 1821) and the South American Nycteliini, Physogasterini, and Praociini with representatives from the Atacama Desert. The Nycteliini, which appear as sister to Praociini and Physogasterini, are represented by the genera Auladera Solier, 1836, Callyntra Solier, 1836, Nyctelia Laterille, 1825, and Psectrascelis Solier, 1836, and the sister taxa thereof, Gyriosomus Guérin-Méneville, 1834 + Pilobalia. While the Physogasterini Philorea Erichson, 1834 + (Physogaster Lacordaire, 1830 + Entomochilus Solier, 1844) occur as monophyletic in our analysis, the Praociini are polyphyletic, with Gyrasida Koch, 1962 as sister to (Praocis Eschscholtz, 1829 + Falsopraocis Kulzer, 1958) + Physogasterini. Antofagapraocis occurs as sister to the latter group. The topology of the second lineage with Pimeliinae from the Atacama Desert shows Caenocrypticoides Kaszab, 1969 (Caenocrypticini, Koch 1958) separated from the rest. The remaining taxa split into a heterogeneous group comprising southern African Zophosini Solier 1834 and Adesmiini Lacordaire, 1859, Mediterranean Erodiini Billberg, 1820, and Tentyriini Eschscholtz 1831 and a branch with the Atacama genera of Edrotini Lacordaire, 1859, Epitragini Blanchard, 1845, Evaniosomini Lacordaire, 1859, Thinobatini Lacordaire, 1859, and Trilobocarini Lacordaire, 1859. Within the latter branch the topology shows with maximum branch support the evaniosomin Melaphorus Guérin-Ménéville, 1834 + Evaniosomus Guérin-Ménéville, 1834 and Aryenis Bates, 1868 as sister to Trilobocara Solier, 1851 (Trilobocarini) and these four taxa appear as sister to the rest of this clade. Within these remaining taxa the epitragin Geoborus Blanchard, 1842 + Nyctopetus Guérin-Ménéville, 1831 (Central Chile) and Salax Guérin-Méneville, 1834 (Trilobocarini) are sister to Achanius, Arthroconus Solier, 1851, Aspidolobus Redtenbacher, 1868, Cordibates, Eremoecus Lacordaire, 1859, Hylithus Guérin-Méneville, 1834, and Thinobatis. While [Hylithus (Edrotini) + Thinobatis (Thinobatini)] + Cordibates (Thinobatini) form a well-supported monophyletic group, the sister group relationships of Achanius (Evaniosomini), Arthroconus (Edrotini), Aspidolobus (Epitragini), and Eremoecus (Trilobocarini) are not fully resolved.

The sister group of Pimeliinae includes in our analyses the subfamilies Lagriinae, Stenochinae, Blaptinae, Alleculinae, Tenebrioninae, and Diaperinae; the latter two being non-monophyletic. The three analyzed taxa of Lagriinae (incl. Adeliini Kirby, 1828, Cossyphini Latreille, 1802, Lagriini Latreille 1825; without representatives in the Atacama Desert) form the sister group to the remaining species of this clade, which in turn is separated into Tenebrio + [Bolitophagus Illiger, 1798 + Eledona Latreille, 1797] from Europe (both Bolitophagini Kirby 1837) and the rest. The latter group contains Tribolium + European Melanimon (Melanimini Seidlitz, 1894) as sister to the remaining taxa. These remaining taxa are further divided into Blaptinae (incl. Blaptini Leach, 1815, Dendarini Mulsant & Rey, 1854, Opatrini Brullé, 1832, Pedinini Eschscholtz, 1829, Platynotini Mulsant & Rey, 1853) with Blapstinus Dejean, 1821 (Opatrini) from the Atacama Desert and a second clade which consists of Alleculinae, Diaperinae, Stenochinae, and several Tenebrioninae. The first branch of that diverse clade separates European Nalassus Mulsant, 1854 (Tenebrioninae: Helopini Latreille 1802) from the rest, which is further separated into Stenochinae (without representatives in the Atacama Desert) and a clade consisting of Alleculinae, Diaperinae, and Tenebrioninae. Within the latter, some members of the polyphyletic Diaperinae (Crypticini Brullé, 1832 and Hypophlaeini Billberg, 1820 from Europe) form together with A. diaperinus (Tenebrioninae) the sister to the rest. The latter clade splits into monophyletic Alleculinae (incl. Alleculini Laporte, 1840, Cteniopodini Solier, 1835) with an undescribed species from the periphery of the Atacama Desert (Alleculinae gen. nov.) and a subclade containing further Diaperinae and Tenebrioninae. The Diaperinae of this subclade, including Phaleria (Phaleriini Blanchard, 1845) from the beaches of the Atacama Desert and Holarctic Diaperini, are sister to the Mediterranean Scaurus Fabricius, 1775 and a clade consisting of Scotobiini Solier, 1838 /Amphidorini LeConte, 1862 from the Atacama Desert and the Neotropical Z. atratus (Tenebrionini). Within the latter group the genus Nycterinus Eschscholtz, 1829 (incertae sedis) is sister to Scotobiini (Ammophorus Guérin-Ménéville, 1830 + [Scotobius Germar, 1824 + Diastoleus Solier, 1838]) and Z. atratus.

Overall, in the neuropeptide tree few branches show low support (Fig. S1). These branches include the position of Achanius to Arthroconus (SH-aLRT = 3.4, UFBoot = 43), Salax as sister to Nyctopetus + Geoborus (SH-aLRT = 14.7, UFBoot = 65), Praocis + Falsopraocis (SH-aLRT = 51.4, UFBoot = 89), Auladera as sister to Callyntra + Psectrascelis (SH-aLRT = 38, UFBoot = 79), Sepidium + Psammetichus (SH-aLRT = 3, UFBoot = 47), Diaperis Geoffroy, 1762 + Neomida Latreille, 1829 (SH-aLRT = 6.9, UFBoot = 86), Nestorinus Guerrero, Vidal & Zúñiga-Reinoso, 2022 + Heliofugus Guérin-Méneville 1831 (SH-aLRT = 28.9, UFBoot = 82), Isomira Mulsant, 1856 as sister to Omophlus Dejean, 1834 + Heliotaurus Mulsant, 1856, and Diaperini/Phaleriini as sister to a clade with Scotobiini/Nycterinus Solier, 1835 /Zophobas Dejean, 1834 + Scaurini Billberg, 1820 (SH-aLRT = 62.9, UFBoot = 91).

All analysed taxa of Alleculinae, Blaptinae, Diaperinae, and Stenochinae, as well as those taxa of Tenebrioninae that nest within the sister clade of Blaptinae (this clade is marked with an arrow in Fig. 2), have a distinct synapomorphy in common, namely an insertion of eight amino acids in the myosuppressin precursor (Fig. 3A; see Data S3 for full sequences). This insertion does not result from differential transcription, but it is indeed manifested at the gene level. This could be verified by genome sequencing of a N. abdominalis specimen and a subsequent comparison of the myosuppressin gene structures (exons) of N. abdominalis and Tr. castaneum (Fig. 3B).

Figure 2 Neuropeptide tree.

BI tree obtained from the analysis of a dataset of 35 peptide precursors from 83 genera of Tenebrionidae (47 tribes, seven subfamilies), including the 30 genera from the Atacama Desert. Assignment of subfamilies and tribes according to Matthews et al. (2010), Bouchard et al. (2021) and Kamiński et al. (2020); Color coding for Tenebrionidae: Alleculinae, yellow; Blaptinae, grey; Diaperinae, light blue; Lagriinae, dark green; Pimeliinae, dark blue; Stenochinae, light green; Tenebrioninae, red. Atacama genera are marked with asterisks. The arrow marks the clade with a synapomorphy in the myosuppressin gene. Posterior probability (PP) and UFBoot (Bt) values are highlighted with circles on the nodes: black, above or equal to 0.95/95; grey, between 0.90−0.94/90-94; white, below 0.90/90. The detailed information on posterior probability/UFBoot values as well as the ML tree are provided in Fig. S1.

Figure 3 Myosuppressin precursor sequences.

Taxon-specific insertion in the myosuppressin precursor sequence, which represents a synapomorphy of a subgroup of Tenebrionidae. (A) Simplified overview of a partial transcript sequence (see Data S3 for full sequences) showing the insertion in genera belonging to different subfamilies (Alleculinae, Blaptinae, Diaperinae, Stenochinae, Tenebrioninae). N. abdominalis position marked with * and Tr. castaneum position marked with +. (B) Part of the corresponding gene sequence of the myosuppressin gene in N. abdominalis (analysed in this study) and the orthologous gene of Tr. castaneum (Li et al., 2008) without that sequence. Color coding: Alleculinae, yellow; Blaptinae, grey; Diaperinae, light blue; Lagriinae, dark green; Pimeliinae, dark blue; Stenochinae, light green; Tenebrioninae, red.

(B) Analysis of a large scale dataset of orthologous genes

The partitioned and concatenated alignment is composed of 1742 OGs with an overall length of 788,676 amino acid sites (Data S2). The best fitting models according to ModelFinder are listed for each partition in Data S2, which also contains the concatenated alignment. The topology of the resulting tree (Fig. 4) is largely congruent with that of the neuropeptide precursor data set. Differences are mainly observed for several of those branches with low support in the neuropeptide precursor tree (see Fig. S1): Salax as sister to a clade comprising Achanius, Arthroconus, Aspidolobus, Cordibates, Eremoecus, Hylithus, and Thinobatis; Praocis as sister to Falsopraocis + Physogasterini; Auladera + Nyctelia as sister to Callyntra + Psectrascelis; Sepidium as sister to Psammetichus + Eurychora; Alleculinae as sister to Scotobiini/Nycterinus/Zophobas + Scaurini; Nestorinus as sister to Heliofugus + Cuphotes Champion, 1887; and Isomira + Prionychus as sister to Omophlus + Heliotaurus. In addition, Discopleurus is sister to a main branch of Pimeliinae (Fig. 4), including, among other tribes, also the Stenosini; and Nycterinus changed its position and was recovered as sister to Diastoleus + Scotobius. In the large scale data set of orthologous genes, the branches with low support (Fig. S2) include that with Zophobas as sister to Nycterinus, Scotobius and Diastoleus (SH-aLRT = 8.2/UFBoot = 61). In both data sets, Corticeus has the same position, but the corresponding branches are very long.

Figure 4 Orthogroups tree.

ML phylogenetic tree obtained from the analysis of a dataset of 1,742 orthogroups from 83 genera of Tenebrionidae, including the 30 genera from the Atacama Desert. Red squares mark species with different positions compared to the neuropeptide tree, the arrow shows the position of the Alleculinae clade as sister to Scotobiini + Scaurini. Color coding and branch support as in Fig. 2. The countries where the taxa were collected are listed after the genus names.

Discussion

Transcriptomic information, mostly obtained from single individuals, was on the one hand used to obtain the amino acid sequences of 35 orthologous peptide precursors of genera of Tenebrionidae from the Atacama Desert and of selected taxa from other regions of the world. Due to their co-evolution with their corresponding receptors, neuropeptide sequences are particularly conserved and very well suited for a reconstruction of phylogenetic relationships at the intra-ordinal level (Bläser, Misof & Predel, 2020; Predel et al., 2012; Roth et al., 2009). Other advantages of using such datasets are the ease of ortholog assignment and the presence of unambiguous and highly conserved sequence motifs that facilitate a manual control of alignments generated by sequence alignment programs. The parallel analysis of the large scale dataset of orthologous genes revealed mostly the same topology as the neuropeptide precursor tree, with the exception of the few differences discussed below. The majority of Atacama genera cluster in three clades. Two of these clades belong to the subfamily Pimeliinae, which contains most of the desert-adapted darkling beetles worldwide (Doyen, 1993; Kergoat et al., 2014b). In the Pimeliinae Pimelia/Akis were found to be the sister group to the remaining 17 analyzed tribes of Pimeliinae. The latter lineage consists of two clades with worldwide distribution, each containing a larger number of Atacama genera. One of these clades contains Nycteliini, Praociini and Physogasterini and forms a well-supported monophyletic group. This confirms previous morphological studies, which suggested Praociini, Physogasterini and Nycteliini as closely related taxa (Doyen, 1972; Doyen, 1993). These tribes are only known from arid regions of South America and are thought to be the sister group of North American Coniontini Waterhouse, 1858, Branchini LeConte, 1862 and Asidini (Doyen, 1993). The Mediterranean Alphasida representing Asidini, was recovered in our analyses as sister to Praociini, Physogasterini and Nycteliini. Different from the most recent cladistic analysis of morphological characters in Nycteliini (Flores, 2000a), our analysis shows monophyletic Nycteliini as sister to Praociini + Phyogasterini. Within Nycteliini, which generally avoid the hyperarid core of the Atacama Desert, Pilobalia + Gyriosomus form the sister clade to the remaining Nycteliini. Physogasterini represent another very well supported monophyletic group in our analyses and include many species typical of the hyperarid core of the Atacama Desert. However, Praociini as defined in Flores (2000b) and Flores & Vidal (2009) appear polyphyletic with both data sets, this tribe requires a re-evaluation based on molecular data. From the four genera included here, Praocis and Falsopraocis are sister to Physogasterini, while Antofagapraocis and the central Chilean Gyrasida do not form a monophyletic group with Praocis and Falsopraocis. The genus Psammetichus, which is typical of hyperarid environments along the Coastal Cordillera of the Atacama Desert and the Pampa de Tamarugal, belongs to a sister clade of the above tribes. That clade also includes Sepidiini and Adelostomini, which do not occur in South America (Bouchard et al., 2021). Kamiński et al. (2022) suggested Sepidiini and Adelostomini as closely related tribes, considering the morphology of female terminalia and several genes. However, they did not place Elenophorini close to these tribes. Psammetichus was transferred to Elenophorini by Doyen & Lawrence (1979), a tribe that also includes Leptoderis (= Elenophorus Dejean 1821) of the western Mediterranean. Leptoderis was also included in our transcriptomic dataset, but the molecular data do not support an ancient link. In fact, Psammetichus was kept in Elenophorini in the past, although it was never found to be closely related to Leptoderis in Doyen’s cladograms (Doyen, 1993). Also Ferrer (2015) doubts this relationship due to a number of morphological characters not shared between Leptoderis and the South American Elenophorini. In our tree, Leptoderis robustly nests within Stenosini, the latter represented by Chilean Discopleurus (within Stenosini only in the neuropeptide tree) and Hexagonochilus, and the palaearctic Dichillus as sister to Leptoderis.

The second major branch of Pimeliinae excl. Akis/Pimelia has Caenocrypticoides as sister to the rest. Caenocrypticoides is a well-established example of members of the same tribe (here Caenocrypticini; Endrödy-Younga, 1996) occurring in widely separated arid regions of Africa and South America, and thus probably representing a relict pattern that points to xerophilic ancestors before the break-up of Gondwana. The sister clade of Caenocrypticoides diverges into one lineage with diverse taxa having a wide distribution in the Palaeartic and Africa, but are not present in South America (Erodiini, Tentyriini, Zophosini, Adesmiini) and a second lineage with South American taxa belonging to Edrotini, Epitragini, Evaniosomini, Thinobatini, and Trilobocarini. The current placement of genera within these tribes is based on morphological characters (e.g., Doyen, 1993; Flores & Aballay, 2015). Although the exact position, particularly those of Arthroconus (Edrotini), Salax (Trilobocarini) and Achanius (Evaniosomini) could not be fully resolved with our data, it is obvious that none of the tribes is monophyletic. This South American clade was already mentioned by Doyen (1993) as a group “not easy to fit with any classification” using morphology and the classification at tribe level of the different genera have seen several changes over time (see e.g., Flores & Aballay, 2015). Doyen (1993) himself suggested transferring Achanius to the Edrotini ( =Eurymetopini Casey, 1907). The first split in this lineage separates Evaniosomus/Melaphorus/Aryenis (Evaniosomini) + Trilobocara (Trilobocarini) from the remaining taxa with maximum branch support. These remaining taxa include, among others, Achanius, Eremoecus, and Salax (Trilobocarini) and thus further genera of the aforementioned tribes and are separated in the neuropeptide tree into Geoborus/Nyctopetus (Epitragini) + Salax and a subclade which, in addition to Achanius, Arthroconus and Eremoecus, also includes Aspidolobus as another representative of the Epitragini. In the large scale dataset of orthologous genes, Salax is sister to all above mentioned taxa, including Geoborus + Nyctopetus. Finally, the well supported sister group relationship of Hylithus (Edrotini) and Thinobatis (Thinobatini) clearly argues against the supposed monophyly of Thinobatini which is only composed of the two genera included in our study (Doyen, 1993; Bouchard et al., 2021).

The sister group of Pimeliinae contains all other tenebrionid taxa analyzed in our study. The basal branching separates Lagriinae from the rest, which shows an early branching of Tenebrio + Bolitophagini and Tribolium + Melanimini. The remaining taxa split into the recently re-established Blaptinae (Kamiński et al., 2020) incl. Blapstinus from the Atacama Desert, and a diverse group of taxa including Stenochinae, Diaperinae, Alleculinae, and Tenebrioninae. Blapstinus appears to be the only tenebrionid genus from the Atacama Desert that has close relatives in North America. The corresponding subtribe Blapstinina Mulsant & Rey, 1853 is in fact restricted to Nearctic and Neotropical regions (Lumen et al., 2020; Kamiński et al., 2022). Monophyly of the analyzed taxa of Lagriinae, Blaptinae, Stenochinae, and Alleculinae was confirmed with maximum branch supports, respectively. On the other hand, polyphyly was evident for Diaperinae and Tenebrioninae (see also, e.g., Gunter et al., 2014; Kergoat et al., 2014b; Kamiński et al., 2020). Most taxa of the darkling beetles currently grouped in the subfamilies Alleculinae, Blaptinae, Diaperinae, Stenochinae, and Tenebrioninae have well-developed hindwings and do not show particular adaptations to hyperarid environments (Doyen, 1993). This does not apply to the Scotobiini, which represent the only endemic tribe of Tenebrioninae in arid South America (Matthews et al., 2010) and include the third cluster of tenebrionid genera in the Atacama Desert. In fact, three of the six genera of Scotobiini (Scotobius, Diastoleus, Ammophorus) inhabit the Atacama Desert and were included in our analysis. Within this clade Scotobius + Diastoleus is sister to Ammophorus in the neuropeptide tree, whereas in the large scale data set of orthologous genes Nycterinus replaces the position of Ammophorus. While the classification within Scotobiini of Diastoleus and the widespread Scotobius has been stable, the systematic position of the genus Ammophorus changed considerably over time. When Solier (1838) established the Scotobiini, he included Ammophorus in this tribe. Shortly afterwards Lacordaire (1859) transferred this genus to Nyctoporini Lacordaire, 1859 (Pimeliinae), where it remained for over 100 years (see, e.g., Kulzer, 1955; Peck, 2006; Peña, 1966b). Later, Vidal & Guerrero (2007) transferred Ammophorus to Elenophorini (Pimeliinae). Based on detailed analyses of morphological characters, Doyen (1993) and Silvestro, Giraldo-Mendoza & Flores (2015) proposed to return the genus to Scotobiini. The result of the neuropeptide tree fits the placement of Ammophorus within Scotobiini based on morphology (Silvestro, Giraldo-Mendoza & Flores, 2015). Also, they share a peculiar synapomorphy with the presence of dome-shaped placoid sensilla on the last segment of the antennae (Doyen, 1993). As sister of Scotobiini appears in the neuropeptide tree Zophobas Dejean, 1834 which is known only from Central and tropical South America (Ferrer, 2011). Nycterinus which is historically listed as the only South American genus within Amphidorini (see Doyen & Lawrence, 1979), belongs to the same monophyletic group in both data sets and was identified as sister to the above mentioned Scotobiini + Zophobas in the neuropeptide tree. Recent molecular phylogeny also showed Nycterinus as not belonging to the North American Amphidorini tribe, but rather to the South American Scotobiine clade which also includes Scotobiini and Zophobas (Johnston et al., 2022). The different results of the two data sets do not yet allow us to determine the specific position for Nycterinus.

The highly scattered appearance of the Tenebrioninae across the phylogenetic tree may question the reliability of our results. However, the topology does not show a mixture of taxa with poorly resolved sister group relationship, nor is it the result from particular poor taxon sampling. With the taxon-specific insertion of eight amino acids into the myosuppressin precursor (see Fig. 3) we have found a distinct synapomorphy at the molecular level clearly supporting Alleculinae, Blaptinae, Diaperinae, Stenochiinae, and a number of Tenebrioninae as a higher level monophyletic group. Based on morphological examinations, Doyen and Tschinkel speculated already in 1982 that Diaperinae, Stenochiinae, and Alleculinae could be derived offshoots of Tenebrioninae. In any case, it does not seem an easy task to redefine any clade as Tenebrioninae, except the one containing Tenebrio and Bolitophagini in our analyses.

Conclusions

Using newly generated transcriptome data, we were able to perform a comprehensive phylogenomic analysis of the tenebrionid fauna of the Atacama Desert and fill a gap in our knowledge of the highly diversified Tenebrionidae. The two datasets used for our analyses show only a few discrepancies that might be resolved by more extensive taxon sampling. The majority of Atacama genera are placed into three groups, two of which belong to typical South American lineages within the Pimeliinae. The suggested very close relationship of Psammetichus with the Mediterranean Leptoderis was not confirmed. Caenocrypticini including the Chilean Caenocrypticoides comprises a small group of genera present in southern Africa and (mostly) the Andean region of South America. These taxa display a combination of characters shared with various clades (Doyen, 1993). Our results provide the first evidence for a position of Caenocrypticoides as the sister of one of the main branches within Pimeliinae. While our data support the monophyly of the Nycteliini, Physogasterini and Scotobiini, this does not hold for the Atacama genera of Edrotini, Epitragini, Evaniosomini, Praociini, Thinobatini, Stenosini, and Trilobocarini. To clarify the relationships of these taxa, it is certainly useful to include more southern South American representatives in future analyses. In general, a detailed systematic revision of each of the latter groups appears necessary. As a side effect of our study, we have found a synapomorphy grouping Alleculinae, Blaptinae, Diaperinae, Stenochinae, and several taxa of Tenebrioninae, but not Tenebrio and Tribolium. This character, an insertion in the myosuppressin gene, defines a higher-level monophyletic group within the Tenebrionidae.

Supplemental Information

Supplemental Information 1 Cross-contamination and statistics of newly sequenced transcriptomes

Click here for additional data file.

Supplemental Information 2 Neuropeptide precursors statistics

Neuropeptide precursors used in this study, including their completeness in the various taxa and the average evolutionary divergence across all sequence pairs in the 91 genera (including outgroup taxa).

Click here for additional data file.

Supplemental Information 3 Neuropeptide Maximum likelihood and Bayesian trees

Phylogenetic trees resulting from BI and ML analyses of the partitioned 34 neuropeptide and neuropeptide–like precursors from 83 genera of Tenebrionidae, including the 30 genera from the Atacama Desert. (A) BI tree with posterior probability values for each branch. (B) ML tree with bootstrap support values for each branch (SH-aLRT test / UFBoot).

Click here for additional data file.

Supplemental Information 4 Maximum likelihood OGs tree

ML tree of the partitioned amino acid supermatrix of 1742 OGs. Each node with branch support values SH-like / UFBoot.

Click here for additional data file.

Supplemental Information 5 Neuropeptide IQ-TREE and BI matrixes

• Matrix for ML analysis presented in Fig. 2 and Fig. S1 (amino acids in PHYLIP format). • Matrix for BI analysis presented inFig. 2 andFig. S1, including partitions and evolutionary models for each partition from ModelFinder (amino acids in NEXUS format). • Partition schemes of IQ-TREE matrix for Fig. 2 and Fig. S1. Available at (DOI: 10.5880/CRC1211DB.35)

Click here for additional data file.

Supplemental Information 6 IQ-TREE matrix and partitions 1742 OGs

- Matrix for ML analysis presented in Fig. 4 and Fig. S2 (amino acids in PHYLIP format).

- Partition schemes of IQ-TREE matrix for Fig. 4 and Fig. S2.

Click here for additional data file.

Supplemental Information 7 Myosuppressin alignment

Alignment with full sequences of the myosuppressin precursor motif shown inFig. 3.

Click here for additional data file.

We thank Gustavo Flores (IADIZA, Mendoza, Argentina) for confirming several species identifications of Atacama tenebrionids, Mario García Paris (Museo Nacional de Ciencias Naturales, Madrid, Spain) for the donation of the Leptoderis specimen, Marcelo Guerrero (Santiago, Chile) for providing several photos that were used to optimize the figures, and Rolf Beutel (Jena, Germany) for useful comments to improve the structure of the manuscript. We also would like to thank Tobias Schulze (Biocenter Cologne) for IT support, Marek Franitza, Christian Becker and Janine Altmüller for transcriptome and genome sequencing (Cologne Center for Genomics), and Peter Heger, Volker Winkelmann and Lech Neiroda for their support in running the analyses at the Regional Computing Centre (CHEOPS) of the University of Cologne.

Additional Information and Declarations

Competing Interests

Author Contributions

Field Study Permissions

DNA Deposition

Data Availability

The authors declare there are no competing interests.

Lapo Ragionieri conceived and designed the experiments, performed the experiments, analyzed the data, prepared figures and/or tables, authored or reviewed drafts of the article, and approved the final draft.

Álvaro Zúñiga-Reinoso conceived and designed the experiments, performed the experiments, analyzed the data, prepared figures and/or tables, authored or reviewed drafts of the article, and approved the final draft.

Marcel Bläser performed the experiments, analyzed the data, prepared figures and/or tables, and approved the final draft.

Reinhard Predel conceived and designed the experiments, analyzed the data, prepared figures and/or tables, authored or reviewed drafts of the article, and approved the final draft.

The following information was supplied relating to field study approvals (i.e., approving body and any reference numbers):

The collecting permits for this study were authorized by the Corporación Nacional Forestal (CONAF, Chile), National Commission on Research, Science and Technology (Namibia), Instituto de Conservação da Natureza e das Forestas (ICNF, Portugal) and Servicio Nacional Forestal y de Fauna Silvestre (SERFOR, Peru) (Collecting permits: CONAF N° 005/2017, 105/2020, 016/2021; CONAF N° 005/2017; 757-758/2021/CAPT; NCRST RPIV01042034; SERFOR Nr D000019-2022).

The following information was supplied regarding the deposition of DNA sequences:

BioProject ID: PRJNA884860.

CRC1211DB DOI: 10.5880/CRC1211DB.35

The following information was supplied regarding data availability:

The data is available at the Collaborative Research Centre 1211 - Database: Ragionieri L, Zúñiga-Reinoso A, Bläser M, Predel R. 2023. Supplementary data of the manuscript entitled Phylogenomics of darkling beetles (Coleoptera: Tenebrionidae) from the Atacama Desert. CRC1211 Database. DOI: 10.5880/CRC1211DB.57.

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
