# Peer review of "Phylogenomics of darkling beetles (Coleoptera: Tenebrionidae) from the Atacama Desert"

_PeerJ, doi:10.7717/peerj.14848_

## Round 0.1 · original submission · Minor Revisions

Dear Dr. Ragionieri and colleagues:

Thanks for submitting your manuscript to PeerJ. I have now received two independent reviews of your work, and as you will see, the reviewers raised some minor concerns about the research. Despite this, these reviewers are optimistic about your work and the potential impact it will lend to research on tenebrionid systematics. Thus, I encourage you to revise your manuscript, accordingly, taking into account all of the concerns raised by the reviewers.

Please note that Reviewer 1 kindly provided a marked-up version of your manuscript.

I look forward to seeing your revision, and thanks again for submitting your work to PeerJ.

Good luck with your revision,

Best,

-joe

·

Basic reporting

A significant and much-needed contribution to the systematics of darkling beetles. The manuscript is (mostly) well-written. I have indicated some minor errors in the attached PDF. The authors are familiar with the background literature, taxonomy, and phylogenetics of Tenebrionidae. Results are (mostly) well-illustrated. I propose to merge figs 1 and 4 into a single one. The manuscript is clear and easy to follow.

Experimental design

Manuscript represent original primary research within the scope of PeerJ. Aside from some minor inconsistencies, the paper is well-structured. In my opinion, the first goal of the paper, as defined in the Introduction, should be dropped or modified, i.e. “insights into the diversification of tenebrionids in the Atacama”. The authors are not discussing this issue in light of their results, which is highlighted by a rather pointless introduction to the Discussion section. Plus, the applied methods are strictly phylogenetic and do not provide any direct insights into the colonization of the Atacama desert by different lineages of darkling beetles (i.e. no ancentral area analysis, no time-calibrate phylogeny).

Validity of the findings

This is one of the first (maybe even first!) contributions to the systematics of darkling beetles that relays on transcriptome analysis. Aside from providing previously unreported insights on the South American taxa, results have an impact on the overall classification/phylogeny of the family. This is especially well-discussed by the authors on many levels (tribes representing different subfamilies). Implemented methods helped to resolve many of the previously existing phylogenetic uncertainties, such as subfamilial affiliation of Helopini (Blaptinae vs Tenebrioninae). To sum it up, conclusions are well stated and linked to the original research question & limited to supporting results.
On the other hand, the topologies presented by the authors are largely consistent with previous research. Assuring that the used dataset and applied methodology can be considered reliable.

Additional comments

I have marked several smaller errors in the attached PDF file.
Neoisocerus ferrugineus represents Dendarini, not Pedinini.

·

Basic reporting

The manuscript is overall very well written and clearly phrased. I am not a native English speaker but I found no obvious linguistic problems or errors.
The list of references seems to be appropriate. I have several comments to the figures:
Fig. 1: the shaded area is very difficult to see. Please use a different color. You might think about including the sample sites in the map to give an idea how well the shaded area was represented.
Fig. 1: please mark Tenebrionidae as a clade in the tree. Please also mark the country where they were collected as you state in the text that in one group the Chile species form a different lineage that those from other countries.
Fig. 3: please include the color code from Fig. 2 also into this figure. The might be printed on different pages which makes it difficult to follow.

Experimental design

Concerning content and methods, the paper is also very straightforward. The analysis are well performed and transparently explained. I added a list with minor comments below.
The paper formulates and adresses the raised questions clearly.

Validity of the findings

The findings of the article are novel and important enough for publication. The conclusions are cleary stated and not overemphasized.

Additional comments

Attached is a list with specific comments:

Line 84: of all phylogenetic reconstructions: this is a rather general statement that implies that this applies to all phylogenetic studies on all animals, which surely was not the intention. I would suggest to replace “all” with “all of these”.
Line 110: “all tenebrionid genera“: change to „all described tenebrionid genera”
Line 152: “Taxonomic determination was carried out by Álvaro Zúñiga-Reinoso and Reinhard Predel”: maybe add the relevant keys as citations?

l. 307-314: The entire paragraph might create the impression that the Attacama species are monophyletic as it states several times that they comprise a lineage. Please resolve this and clearly state that these lineages not only comprise Attacama species.

l. 313: here you state that the Chile genera fall into a different lineage than those from other countries. Do those species only occur in Chile or did you only collect them there? The same applies to those from other countries?

Result section: this part is extremely difficult to read and follow due to the numerous mentionings of species and other taxonomic ranks including their authors. I would suggest to name the authors in the table and M&M section but skip them here for the sake of readability.

Paragraph 352: please start this paragraph by stating which subfamilies are monophyletic and which ones not. The non-monophyletic ones are usually stated in “ ”.

Line 364: the sentence starting with “Members of Diaperinae“ hast to begin with „Within the latter,“. Additionally it creates the impression that this includes all Diaperinae which is not true. I would change it to “some members of the polyphyletic Diaperinae”

l. 386: please indicate the position of this synapomorphy in the tree figure.

l. 416: “our focus area” is not a good wording. Please rephrase.

Paragraph l. 411: it is not really clear what the current manuscript can provide to the discussion of this paragraph. At the moment it reads like a review.

l. 563: Nevertheless, it does not seem an easy task to redefine any clade as Tenebrioninae except that which includes Tenebrio and Bolitopagini in our analyses: I do not understand this sentence.

l.571: here is a word in the second half of the sentence missing.

l. 572: please name the two lineages.

Please provide a table (and a map?) with the geographic locations of all sequenced specimens. In relation to this, the ms does not address the question how well the Atacama desert was sampled for the project. The number of 30 (out of 34) genera is very impressive and certainly required a tremendous effort in collecting and travelling. However, it would be interesting to discuss the geographic distribution (and its potential bias) in the taxon sampling.

---

## Round 0.2 · accepted · Accept

Dear Dr. Ragionieri and colleagues:

Thanks for revising your manuscript based on the concerns raised by the reviewers. I now believe that your manuscript is suitable for publication. Congratulations! I look forward to seeing this work in print, and I anticipate it being an important resource for groups studying tenebrionid systematics. Thanks again for choosing PeerJ to publish such important work.

Best,

-joe